# Wheat DOF transcription factors TaSAD and WPBF regulate glutenin gene expression in cooperation with SPA

**Marielle Merlino**[1], **Jean-Charles Gaudin**[2], **Mireille Dardevet**[1], **Pierre Martre**[3], **Catherine Ravel**[1☯], **Julie Boudet**[1☯]*

1 INRAE, Clermont Auvergne University, UMR GDEC, Clermont-Ferrand, France, 2 INRAE, ONF, UMR0588 BioForA, Paris, France, 3 LEPSE, Univ. Montpellier, INRAE, Institut Agro Montpellier, Montpellier, France

☯ These authors contributed equally to this work.
* julie.boudet@uca.fr

**Data Availability Statement:** All relevant data are within the paper and its Supporting Information files.

## Abstract

Grain storage proteins (GSPs) quantity and composition determine the end-use value of wheat flour. GSPs consists of low-molecular-weight glutenins (LMW-GS), high-molecular-weight glutenins (HMW-GS) and gliadins. GSP gene expression is controlled by a complex network of DNA-protein and protein-protein interactions, which coordinate the tissue-specific protein expression during grain development. The regulatory network has been most extensively studied in barley, particularly the two transcription factors (TFs) of the DNA binding with One Finger (DOF) family, barley Prolamin-box Binding Factor (BPBF) and Scutellum and Aleurone-expressed DOF (SAD). They activate hordein synthesis by binding to the Prolamin box, a motif in the hordein promoter. The BPBF ortholog previously identified in wheat, WPBF, has a transcriptional activity in expression of some GSP genes. Here, the wheat ortholog of SAD, named TaSAD, was identified. The binding of TaSAD to GSP gene promoter sequences *in vitro* and its transcriptional activity *in vivo* were investigated. In electrophoretic mobility shift assays, recombinant TaSAD and WPBF proteins bound to cis-motifs like those located on HMW-GS and LMW-GS gene promoters known to bind DOF TFs. We showed by transient expression assays in wheat endosperms that TaSAD and WPBF activate GSP gene expression. Moreover, co-bombardment of Storage Protein Activator (SPA) with WPBF or TaSAD had an additive effect on the expression of GSP genes, possibly through conserved cooperative protein-protein interactions.

## Introduction

Bread wheat (*Triticum aestivum* L.) is one of the most important crops worldwide, with over 660 million tons harvested in 2020 (http://faostat.fao.org/). Wheat provides on average 22% of the total calories and protein in the human diet [1]. Wheat flour is suitable for bread making because wheat grain storage proteins (GSPs), the main components of gluten, give dough unique viscoelastic properties. Gliadins are monomeric GSPs, while glutenins are polymeric

**Funding:** This work was supported by the French National Research Agency (ANR) and France AgriMer in the framework of the Investments for the Future BreedWheat project (ANR-10-BTBR-03). No, the funders had no role in study design, data collection and analysis, decision to publish, or preparation of the manuscript.

**Competing interests:** The authors have declared that no competing interests exist.

GSPs composed of low-molecular-weight glutenin subunits (LMW-GS) and high-molecular-weight glutenin subunits (HMW-GS) [2, 3]. Gliadins contribute to the viscous properties of wheat dough by conferring extensibility, while glutenins play a key role in strengthening it by conferring elasticity [4]. The GSP genes are located on the homoeologous loci of chromosome groups 1 and 6. Particularly, HMW-GS genes are located on the long arms of chromosomes 1A, 1B and 1D at loci *Glu-A1*, *Glu-B1* and *Glu-D1* respectively [5]. Each locus consists of two closely linked paralogous genes, *Glu-1-1* and *Glu-1-2* that encode x-type and y-type HMW-GS, respectively. LMW-GS genes are located at *Glu-A3*, *Glu-B3*, *Glu-B2*, *Glu-B4* and *Glu-D3* loci on the short arms of chromosomes 1A, 1B and 1D, respectively [6]. The amounts of each GSP in mature grain determines the nutritional and rheological quality of the wheat flour.

GSP synthesis in developing grain is determined by nitrogen and sulfur availability and is finely controlled at a transcriptional level [7, 8]. A conserved regulatory network of DNA-protein and protein-protein interactions activates the transcription of GSP genes, and is largely described in barley [9–11] with evidence from other cereals [12–15] and dicots [16, 17]. In wheat, conserved cis-regulatory motifs have been identified in the promoters of genes encoding GSPs, and a number of transcription factors (TFs) involved in GSP synthesis regulation have been characterized. For example, Storage Protein Activator (SPA) and SPA Heterodimerizing Protein (SHP) are two wheat bZIP-type TFs that bind to GCN4-like motifs (GLM) and G-box motifs in the promoters of genes encoding glutenins. SPA activates transcription of these genes [18, 19] while SHP represses it [19]. Wheat Prolamin Box Binding Factor (WPBF) is a DNA-binding with One Finger (DOF) TF that binds to the Prolamin-box (P-box) or Endosperm Motif (EM) sequences in the promoters of genes encoding HMW-GS and LMW-GS [20–22]. The WPBF encoded by the D genome binds *in vitro* to the P-box in the promoters of *Glu-B1-2* and *Glu-D1-1* [22]. Previously, Conlan et al. [21] have suggested that WPBF does not have a regulatory activity but rather mediates the regulatory activity of SPA. By contrast, Dong et al. [23] have shown that WPBF activates an α-gliadin gene. A B3-type TF called TaFUSCA3 activates *Glu-B1-1* by binding to the RY-box motif in its promoter [24]. In addition, a TF of the R2R3MYB family, TaGAMYB, activates HMW-GS gene expression by directly binding to the 5′ -AACA/TA-3′ cis-motif [25]. This basic network has been enriched by information about TFs of the NAC family (No Apical Meristem–ATAF1 –CUC2 (Cup-shaped Cotyledons)). This is one of the largest groups of plant-specific TFs involved in developmental and growth processes [26]. To date, functional characterizations have confirmed the regulatory activity of four NAC TFs in GSP synthesis [27–30].

The well-characterized DOF family includes plant-specific TFs involved in numerous physiological processes of plant growth and development [31–33]. As DOF TFs are involved directly in the regulation of GSP synthesis they also participate to the carbon and nitrogen assimilation regulation [34–37] and carbohydrate metabolism [38–41]. DOF proteins generally comprise 200–400 amino acids and are mainly characterized by the presence of an N-terminal DOF domain and a C-terminal transcriptional regulation domain [42]. The highly conserved DOF domain consists of 52 amino acids including a Cys2/Cys2 zinc finger motif with the sequence $Cx_2C-x_{21}-Cx_2C$. This motif binds to zinc ($Zn^{2+}$) in a characteristic zinc finger configuration. Replacing the conserved cysteine residues with serine abolished the DNA-binding capacity of the DOF domain [43]. The DOF domain is essential for recognizing the core sequence 5′ -(AT)/AAAG-3′ (or the reverse complementary sequence CTTT) in genomic regions upstream of the target genes. Diversity in the amino acid sequences of the transcriptional regulation domains is expected to reflect the various functions of the DOF proteins. Two DOF TFs bind the P-box motif, barley PBF (BPBF, WPBF ortholog) and Scutellum and Aleurone-expressed DOF (SAD) [43–45]. According to transient expression experiments in developing barley endosperms, SAD trans-activates transcription from the promoter of

*Hor2* which encodes a hordein, a barley GSP. When the two DOF factors BPBF and SAD were co-bombarded into developing barley endosperms, an additive effect on transcription activation was observed [45]. Originally considered only as a DNA-binding domain, now the DOF domain is regarded as a bifunctional domain with both DNA-binding and protein-protein interaction activities [42]. Indeed, there have been several demonstrations that DOF and bZIP proteins interact with each other. For example, maize PBF interacts with the Opaque2 bZIP protein [46]. Rice PBF (RPBF), an ortholog of BPBF, interacts with the RISBZ1 bZIP TF in inducing the expression of the rice GSP genes [47]. Bioinformatic analysis revealed high binding activity for the EcDof-EcO2 heterodimer onto GSP promoters in finger millet [48]. DOF proteins are also able to interact with MYB TF. In barley, BPBF and SAD interact with a barley GAMYB, which itself binds a 5′-AACAAC-3′ element, close to the EM. Cooperatively the TFs induce expression of the *Hor2* gene [45, 49].

To date, while PBF orthologs have been identified in maize, rice and wheat, no ortholog of barley SAD has been found in other cereals [43, 46, 47]. The conservation of molecular actors involved in GSP regulation between cereals suggests that another DOF TF binds the wheat P-box. Sequence homology analysis based on the barley SAD sequence allowed us to find this TF in wheat. The TaSAD function in GSP synthesis was studied *in vitro* and *in vivo*. Electrophoretic mobility shift assays (EMSA) and transient expression assays on immature wheat endosperms were performed to determine its DNA binding and regulatory activities on glutenin gene expression. TaSAD function was also characterized in combination with WPBF and SPA proteins.

## Materials and methods

The wheat genome is made of three homoeologous subgenomes (A, B and D), so most genes are present in three copies. Here, specific copies are referred to by an upper-case suffix indicating the corresponding sub-genome.

### Identification of a wheat ortholog of barley SAD

The nucleotide sequence AJ312297.1 of *SAD* from *Hordeum vulgare* was used to identify wheat (*Triticum aestivum*) homologs by a Blast search on the wheat data library (https://plants.ensembl.org/). The DOF domains of three homoeologous TaSAD sequences were aligned with SAD using ClustalW with default parameters [50]. In addition, the amino acid sequence of TaSAD-B and 14 other DOF protein sequences were retrieved from literature (S1 Table) and aligned with ClustalW. Based on this alignment, a phylogenetic tree was built with MEGA 7 [51] using the UPGMA method based on a Jones-Taylor-Thornton model matrix-based model. The rate of variation among nucleotide sites was modeled with a gamma distribution (shape parameter = 4) and 1,000 bootstrap samplings were made. The bootstrap consensus tree is shown.

### Gene expression analysis

To quantify gene expression, RNA was extracted from developing wheat grains (cultivar NB1), harvested every 100˚Cdays between 300 and 700˚C days after anthesis (four independent replicates). Transcript levels of three housekeeping genes [β-tubulin, glyceraldehyde 3-phosphate dehydrogenase (GAPDH), and elongation factor 1 alpha (eF1α)] and *TaSAD*, *WPBF*, *SPA* were quantified by real-time qPCR using Lightcycler 480 SYBR Green I Master (Roche, http://www.roche.com/) as described in Boudet et al. [19]. The primers defined for *TaSAD*, *WPBF*, and *SPA* simultaneously amplified the three homeologs of the respective genes. The primers

for HMW-GS amplified the four genes expressed in NB1 while those for LMW-GS amplified several members of this gene family. The sequences of the primers used are given in S2 Table.

## Cloning, expression and purification of recombinant wheat proteins in *Escherichia coli*

Full-length *WPBF* cDNA was synthetized using as a template the D copy of Chinese Spring wheat variety [52] while *TaSAD* full-length cDNA was synthetized based on the B copy of Renan variety deduced from the genomic sequence obtained from the BAC library described in Chalhoub et al. [53].

   To produce recombinant TaSAD protein, *TaSAD-B* cDNA was inserted into the pET32-TEV plasmid (Novagen, Merck) between the *Bam*HI and *Hind*III sites as explained by Boudet et al. [19]. The recombinant protein produced (named Trx-TaSAD) consists of an N-terminal thioredoxin (Trx) and six histidine residues fused with TaSAD. The recombinant protein expression in *E. coli* BL21-DE3 strain (Invitrogen, Life Technologies) and purification was described in Boudet *et al.* [19]. The same procedure was used to express and produce the recombinant Trx-WPBF protein, using the *WPBF-D* cDNA. S1 Fig shows the results of expression and purification of these two recombinant proteins. A recombinant SPA protein, produced as in Ravel et al. [54], was also used in EMSA experiments.

## Electrophoretic mobility shift assay (EMSA)

Eight putative DOF binding cis-motifs described in previous studies were synthetized by Sigma (Saint-Louis, Missouri) (S3 Table). The motifs include EM1 and EM2 from *GluD3* promoter [18], DOF1, DOF2, DOF3 from *GluB1-1* promoter [54], and Pb2 from *GluD1-1* promoter [55]. The other two cis-motifs used were Pb1-like and Pb3-like (S3 Table), which are respectively similar to the P-boxes Pb1 and Pb3 described by Norre et al. [55]. Mutated versions of EM and DOF cis-motifs were also synthesized by Sigma (S3 Table).

   Single-stranded DNA probes were labelled and hybridized as described in Boudet et al. [19] using the biotin 3' End DNA Labeling Kit (Pierce). DNA-protein binding reactions (20 fmol of labelled dsDNA probes, 370 ng of Trx-TaSAD, 500 ng of Trx-WPBF and 625 ng of His-SPA), competitions with unlabeled probes (100× and 200× the amount of unlabeled DOF or EM probes) or mutated probes, and subsequent analysis of DNA-protein complexes were performed as described in Boudet et al. [19].

## Transient expression assays in wheat endosperm

Reporter constructs pGluB1-1 and pGluD3 previously analyzed by Boudet et al. [19] were used for particle bombardment of wheat endosperm. These constructs were obtained using Gateway technology (Invitrogen) to clone promoters with a β-glucuronidase (*GUS*) reporter gene. As effectors, the complete *TaSAD-B* and *WPBF-D* cDNAs were each placed under the control of the maize *Ubiquitin* promoter plus the first intron of the *Ubiquitin* gene using Gateway technology (Invitrogen) as described in Boudet et al. [19], to give pUbi-TaSAD and pUbi-WPBF. The pUbi-SPA effector and the pUbi-GFP (Green Fluorescent Protein) control bombardment efficiency construct were also used [19].

   Seeds of *T. aestivum* cv. Récital were grown into polyvinyl chloride (PVC) columns (volume 1.9 L, inner diameter 7.5 cm, length 50 cm, three plants per column), filled with a 2/3:1/3 mixture of potting soil and perlite, in a controlled growth chamber at 20°C/16°C (day/night) with 16 h photoperiod. Fifteen days after sowing and for the following four weeks, plants received a 68 mL per day of a 3 mmol N $L^{-1}$ nutrient solution containing 1 mM $KH_2PO_4$, 1 mM $KNO_3$, 0.5 mM $Ca(NO_3)_2$, 0.5 mM $NH_4NO_3$, 0.1 mM $MgSO_4$, 2 mM $MgCl_2$, 3.5 mM $CaCl_2$, and 4

mM KCl to provide macroelements, and 10 μM $H_3BO_3$, 0.7 μM $ZnCl_2$, 0.4 μM $CuCl_2$, 4.5 μM $MnCl_2$, 0.22 μM $MoO_3$, and 50 μM EDFS-Fe to provide microelements. After heading, plants received water only. Endosperms were collected 220°C days after anthesis.

Transient transformation was performed by bombarding endosperms with gold particles coated with plasmid according to Boudet et al. [19]. After bombardment, endosperms were incubated for 24 h in the dark at 30°C in a Murashige and Skoog medium supplemented with 3% (w/v) sucrose. *GUS* and *GFP* expression were quantified according to Boudet et al. [19]. Fluorescence from pUbi-GFP expression was used to determine the efficiency of bombardment and to normalize *GUS* expression (the number of GUS foci divided by the number of GFP foci). For each combination of constructs, three or four independent bombardments of three Petri dishes containing eight endosperms each were performed.

## Statistical analysis

Results of the transient expression assays were analyzed by ANOVA with reporter/effector construct as the factor and normalized GUS expression as the variable followed by a multiple pairwise comparisons post-hoc Tukey's test. Tukey's test was also used to compare the means of normalized GUS expression values obtained with different combinations of reporters and effectors. Statistical differences were judged at the 0.05 confidence level.

## Results

### Wheat orthologs of barley *SAD* are expressed during grain filling

Two DOF TFs, BPBF and SAD, that activate the transcription of hordein genes in barley have been found [43–45]. Only the BPBF ortholog WPBF has been identified in wheat [20–22]. This led us to search for the wheat orthologs of SAD. The barley *SAD* gene sequence was used in a Blast search of the *Triticum aestivum* genome (https://plants.ensembl.org/) with the result that three homoeologous sequences located on the group 6 chromosomes were retrieved. These *Triticum aestivum SAD* (*TaSAD*) genes contain two exons and a single intron (Fig 1A). The characteristic DNA-binding domain of the DOF family is located at the beginning of exon2. The coding sequences (CDS) of the A, B and D *TaSAD* copies encode polypeptides of 392, 389 and 391 amino acid residues, respectively. TaSAD-B protein has the highest identity (91.26%) with barley SAD compared to 91.07% for TaSAD-A and 90.79% for TaSAD-D. The DNA-binding domains of the four proteins (the three from wheat and one from barley) are identical (Fig 1B). This conserved domain of 54 amino acids contains four cysteine residues for putative zinc coordination. The CDS of the B genome copy (referred to simply as *TaSAD* hereafter) isolated from the wheat cv. Renan showed 99.06% identity with the B genome copy (TraesCS6B02G270100) from cv. Chinese Spring. Their respective protein sequences showed 98.45% identity.

To investigate the evolutionary relationships between TaSAD and SAD, a phylogenetic tree was generated with several other DOF protein sequences (Fig 1C). In this tree, we observed a first group made of three subgroups. In the first subgroup, TaSAD was grouped with SAD and with OsDof2 and OsDof1 [56] which are probably their rice orthologues. The second subgroup contains three *Arabidopsis thaliana* proteins, AtDOF4-6, 3–7 and 2–5, involved in the control of seed germination [57, 58]. OsDof3 (also known as RPBF) which is involved in gene regulation of rice seed storage proteins [47], forms the third subgroup with ZmPBF, BPBF and WPBF associated with endosperm development [43, 46]. The last group included four proteins, ZmDof3, the paralogs ZmDof1 and ZmDof2 and OsDof4. ZmDof3 is involved in starch accumulation and aleurone development in maize endosperm [40]. The paralogs ZmDof1 and

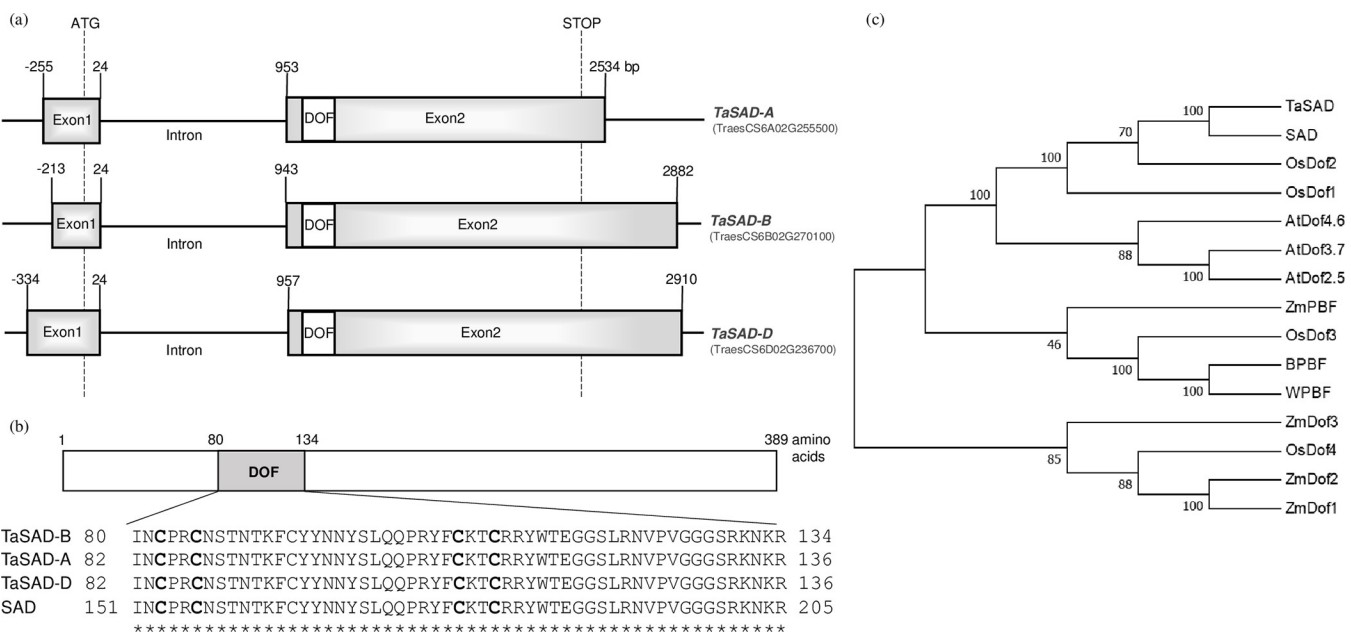

**Fig 1. Genomic structure of the wheat *TaSAD* genes, multiple sequence alignment of DOF domain and phylogenetic tree analysis of TaSAD and 14 other annotated DOF proteins.** (a) Schematic representation of the three homoeologous genes (*TaSAD-A*, *B*, and *D*) encoding TaSAD from Chinese Spring showing the two exons separated by a single intron. The position of the DOF domain is indicated as a white box. The translation initiation ATG and the STOP codon positions are indicated. Nucleotide positions are given relative to the start codon. (b) Multiple amino acid sequence alignment of TaSAD-B, -A, -D, and barley SAD DOF domains. Asterisks indicate conserved amino acids in all four sequences. The positions of the cysteine residues for putative coordination of zinc are indicated in bold. (c) Bootstrap consensus tree drawn with the UPGMA method. The percentage of replicate trees in which the associated taxa clustered together in the bootstrap test (1000 replicates) are shown next to the branches. Ta or W, *Triticum aestivum;* B, *Hordeum vulgare;* Os, *Oryza sativa;* At, *Arabidopsis thaliana;* Zm, *Zea mays*. Details of these DOF proteins (name, species, Genbank number and references) are listed in S1 Table.

ZmDof2 are involved in tissue-specific and light-regulated gene expression [59], and OsDof4 plays important roles in regulating rice flowering time [60].

Relative expression of *TaSAD*, W*PBF* and *SPA* was measured by qRT-PCR in developing endosperms of the bread wheat genotype NB1 (Fig 2A). *TaSAD* expression was low compared with *WPBF* and *SPA*, and varied very little during grain filling. W*PBF* and *SPA* transcript levels increased gradually from 300 to 600˚C days after anthesis then decreased, with the *WPBF* expression level returning to its initial level at 700˚C days after anthesis. *WPBF* was the most expressed of the three-studied TFs. *TaSAD* expression was 50 to 200 times lower than *WPBF* expression and 70 to 100 times lower than *SPA* expression. Expression levels of *HMW-GS* and *LMW-GS* genes increased gradually from 300 to 600˚C days and then decreased, similar to the changes in *WPBF* and *SPA* expression (Fig 2B).

## DNA binding activities of TaSAD and WPBF

Norre et al. [55] described three prolamin-box motifs (Pb1, Pb2, and Pb3) in the 5'-proximal region of the *GluD1.1*. Pb2 is conserved in the *GluB1-1* gene promoter, whereas only partial Pb1 and Pb3 motifs are retrieved, named Pb1-like and Pb3-like (S2A Fig). Ravel et al. [54] annotated three DOF cis-motifs (DOF1, DOF2 and DOF3) in the *GluB1-1* promoter. To test whether TaSAD and WPBF recognize the Pb, Pb-like and/or DOF cis-motifs of the *GluB1-1* promoter, synthetic oligonucleotides containing these motifs were tested by EMSA with WPBF and TaSAD recombinant proteins. No mobility shifts were observed with Pb2, Pb1-like or Pb3-like motifs (data with Pb3-like are shown in S2B Fig as an example). In contrast, WPBF and TaSAD did bind each of the three DOF motifs *in vitro* (Fig 3A). For both Trx-WPBF and

(a)

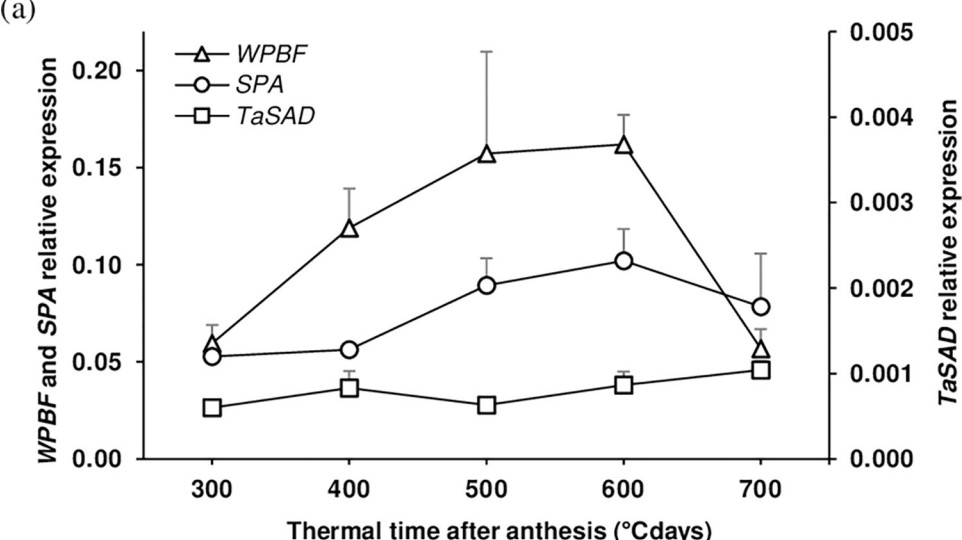

(b)

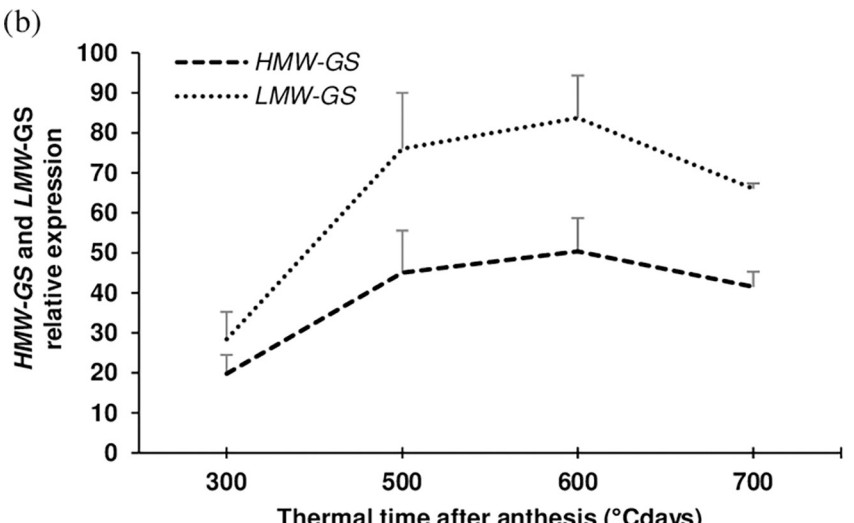

**Fig 2.** Quantitative RT-PCR measurements of expression of (a) *TaSAD*, *WPBF*, *SPA*, (b) *HMW-GS* and *LMW-GS* in grains of the wheat cultivar NB1 from 300 to 700 degree days after anthesis (˚Cdays). Quantitative RT-PCR was performed to quantify the combined expression of the three homoeologous copies of each of *TaSAD*, *WPBF*, and *SPA*, the four *HMW-GS* genes and several *LMW-GS* genes. Data are means +SD for n = 4 independent replicates. Quantitative RT-PCR measurements of three housekeeping genes were used to estimate relative transcript levels.

Trx-TaSAD recombinant proteins, shifted bands of DNA-protein complexes were clearly observed with DOF2 and DOF3 motifs, while the DOF1 shift was much fainter. The binding specificity of the recombinant proteins was tested by adding a molar excess of competing unlabeled probes to the reaction, which had the effect of diminishing all retarded bands. The mutation of one or two nucleotides in the core sequence AAAG of probes, as in *dof1*, *dof2* and *dof3*, did abolish the Trx-WPBF and Trx-TaSAD bands (Fig 3A). EMSA was also performed with the two recombinant proteins and probes containing the EM motifs from the *GluD3* promoter (Fig 3B) reported to bind WPBF [20, 21]. EM1 and EM2 specifically produce retarded bands when incubated with Trx-WPBF or Trx-TaSAD recombinant proteins. These interactions were abolished when the AAAG motif was mutated in *em1* and *em2* (Fig 3B). These bindings

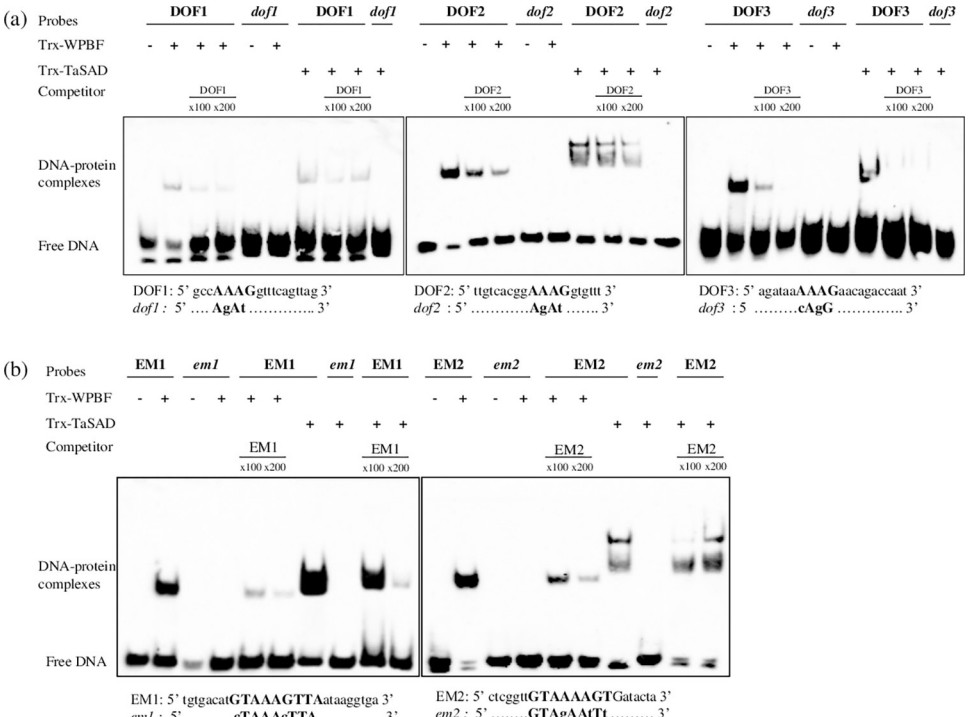

**Fig 3. Electrophoretic mobility shift assays (EMSA) of recombinant WPBF and TaSAD proteins with oligonucleotide sequences equivalent to sequences found in *GluB1-1* (DOF1, DOF2 and DOF3) and *GluD3* (EM1 and EM2) promoters.** (a) EMSA of recombinant Trx-WPBF (500 ng) and Trx-TaSAD (370 ng) proteins with the 20 bp biotin-labeled DOF1 (-742bp), DOF2 (-713bp) and DOF3 (-345bp) probes based on sequences in the *GluB1-1* gene promoter and their mutated versions *dof1*, *dof2* and *dof3*. (b) EMSA of recombinant Trx-WPBF (500 ng) and Trx-TaSAD (370 ng) proteins with the 26 bp and 22 bp biotin-labeled EM1 (-303bp) and EM2 (-337bp) probes based on sequences in the *GluD3* gene promoter and their mutated versions *em1* and *em2*. Twenty fmol of the probes were used. Competition experiments were performed by using 100x and 200x of the unlabeled probes. All sequences of the oligonucleotides used as probes are shown with the cis-motifs in bold; identical residues are represented by dots and mutated nucleotides are shown in lowercase.

were competed by a molar excess of the unlabeled intact probes. We noticed that two DNA-protein complexes and a larger retarded band were observed when TaSAD was combined with DOF2, EM2 and EM1 motifs, respectively. In summary, the results revealed that *in vitro* WPBF and TaSAD specifically bound to the three DOF motifs in the *GluB1-1* promoter and the two EM motifs in the *GluD3* promoter.

## TaSAD and WPBF regulate the transcription of *GluB1-1* and *GluD3*

The functional relevance of the *in vitro* interactions observed between WPBF or TaSAD and the DOF and EM cis-motifs was tested *in vivo* by transient expression assays in wheat endosperms. Fig 4A shows the reporter constructs with the *GluB1-1* or *GluD3* promoters used in the assays (pGluB1-1 and pGluD3, respectively). Immature endosperms were transiently transformed by particle bombardment with the reporter alone or in combination with the effector constructs in a 1:1 molar ratio. When pGluB1-1 was co-transfected with pUbi-TaSAD or pUbi-WPBF, the GUS activity was significantly higher ($P < 0.001$) than in transfections with pGluB1-1 alone (Fig 4B). GUS activity was significantly higher with pUbi-WPBF (normalized GUS expression 0.45) than with pUbi-TaSAD (0.38). Co-transfection of pGluB1-1 with pUbi-WPBF and pUbi-TaSAD effectors resulted in similar levels of GUS activity to the levels induced by the combination of pGluB1-1 and pUbi-WPBF ($P = 0.404$).

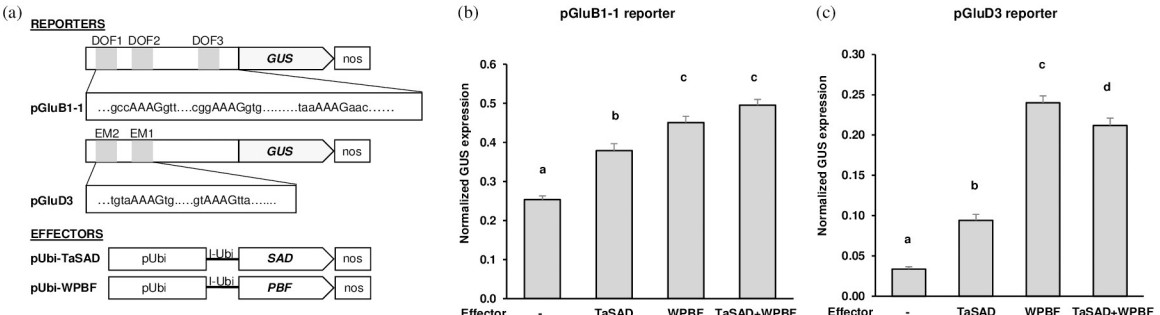

**Fig 4. Assays of transient expression from the *GluB1-1* and the *GluD3* promoters in the presence of TaSAD and WPBF in developing wheat endosperms.** (a) Schematic representation of the reporter and the effector constructs. The reporter constructs consisted of the *uidA* reporter gene (GUS) driven by 747 bp of the *GluB1-1* promoter (pGluB1-1) or by the 349 bp of the *GluD3* promoter (pGluD3). The nucleotide sequences of the DOF and EM motifs are also shown. The effector constructs contained the complete cDNA of *WPBF* or *TaSAD* under the control of the Ubiquitin promoter (pUbi) followed by the first intron of the *Ubi* gene (I-Ubi), and were flanked downstream by the 3' nos terminator (nos). (b) (c) Transient expression assays after co-bombardment of developing wheat endosperms with the pGluB1-1 or the pGluD3 reporters and the different effectors in equimolar proportions. The pUbi-GFP construct, containing the cDNA of *GFP* under the control of the Ubiquitin promoter (pUbi), was used to determine the efficiency of bombardment. GUS activity was evaluated as the number of blue spots counted per bombarded endosperm and normalized by dividing the number of GUS foci by the number of GFP foci. The means (+ SE) of normalized GUS expression per bombarded endosperm are presented as bar charts for at least 3 independent bombardment events. The number of bombarded endosperms per treatment varied between 52 and 149. Letters a, b, and c and d above the bar indicate significant differences between multiple pairwise comparisons of reporter/effector combinations from an ANOVA followed by the Tukey's post-hoc test.

When pGluD3 was co-transfected with the pUbi-TaSAD or the pUbi-WPBF effectors, GUS activity increased ($P < 0.001$) 3- and 8-folds compared with endosperm transfected with pGluD3 without effectors, respectively (Fig 4C). The 1:1 mixture of pUbi-WPBF and pUbi-TaSAD effectors with pGluD3 also showed significantly higher GUS activity compared with pGluD3 alone ($P < 0.001$). While this difference was higher than co-transfection of pGluD3 with the pUbi-TaSAD it was less than co-transfection of pGluD3 with pUbi-WPBF. Thus, our results show that TaSAD and WPBF are activators of *GluB1-1* and *GluD3* expression.

## Involvement of SPA in the regulatory activity of TaSAD and WPBF

As DOF and bZIP proteins are known to interact with each other to regulate GSP expression [46, 47], the possibility that TaSAD, WPBF and SPA transactivate the *GluB1-1* and *GluD3* promoters was considered and investigated in endosperms (Fig 5). Although SPA has already been described as an activator of *GluB1-1* expression [19], in our conditions, no significant increase in expression was observed with the SPA effector and pGluB1-1 reporter relative to the reporter alone (Fig 5A). Adding pUbi-SPA in a co-transfection of pGluB1-1 with pUbi-TaSAD increased GUS activity 2.5-folds ($P < 0.001$). A similar level of induction was observed when pGluB1-1 was co-transfected with pUbi-SPA and pUbi-WPBF effectors. This expression activation of *GluB1-1* in the presence of pUbi-SPA (normalized GUS expression of 0.78 with pUbi-TaSAD and of 0.72 with pUbi-WPBF) was significantly higher than that observed with pUbi-TaSAD or pUbi-WPBF alone (0.38 and 0.45, respectively) or together (0.50; Fig 4B) showing that SPA participated in activating *GluB1-1* expression.

As for *GluB1-1*, SPA has been described as an activator of *GluD3* expression, and here it induced a 4-folds increase in GUS activity (Fig 5B). When pGluD3 was co-transfected with pUbi-SPA and either pUbi-TaSAD or pUbi-WPBF, GUS activity was 2- and 3.3-folds higher ($P < 0.001$) than the activity induced by pGluD3 with pUbi-SPA, respectively. Activation of *GluD3* expression from pGluD3 co-transfected with pUbi-SPA and pUbi-TaSAD effectors was not different (normalized GUS expression 0.21, $P = 1.000$) as that obtained with pUbi-WPBF

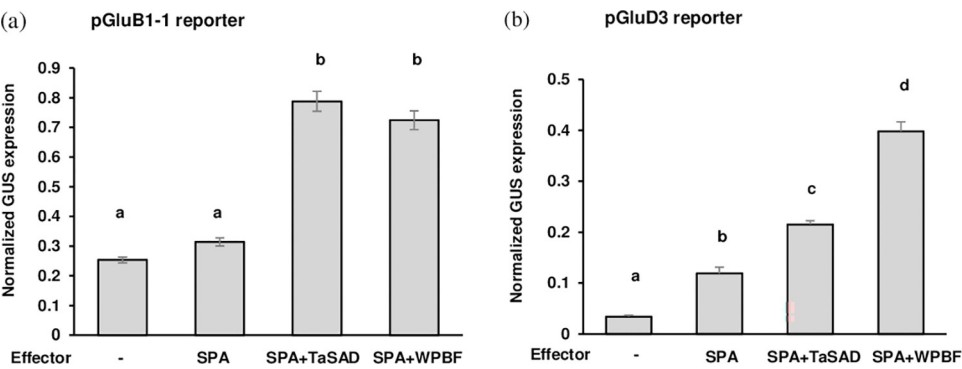

**Fig 5.** Assays of transient expression from the *GluB1-1* (a) and the *GluD3* (b) promoters in the presence of SPA with TaSAD or WPBF in wheat developing endosperms. Transient expression assays involved co-bombardment of developing wheat endosperms with the pGluB1-1 or the pGluD3 reporters and the different effectors in equimolar proportions. The reporter and effector constructs were those described in Fig 4. An additional effector was used, pUbi-SPA which consisted of the complete cDNA of SPA under the control of the Ubiquitin promoter (pUbi), followed by the first intron of the *Ubi* gene (I-Ubi) flanked downstream by the 3' nos terminator (as shown in the schematic representation of the effector constructs in Fig 4(A)). GUS activity was evaluated as in Fig 4. The mean (+ SE) of normalized GUS expression per bombarded endosperm from the reporter alone is that obtained in Fig 4. The number of bombarded endosperms varied per treatment between 37 and 149. Letters a, b, c and d above bars indicate significant differences between multiple pairwise comparisons of reporter/effector combinations from an ANOVA followed by the Tukey's post-hoc test.

and pUbi-TaSAD (Figs 4C and 5B). However, the activation was significantly higher with pUbi-SPA and pUbi-WPBF (0.40) than with pUbi-WPBF and pUbi-TaSAD (0.21) or with pUbi-WPBF alone (0.24; Fig 4C). Once again, it appears that SPA participates in the activation of the *GluD3* gene, which is stronger through association with WPBF than with TaSAD.

To further investigate the cooperation of SPA with TaSAD and WPBF, the binding activity of TaSAD and WPBF in the presence of SPA was analyzed by EMSA. The EM1 and DOF2 probes used were derived from the *GluD3* and *GluB1-1* promoters respectively (Fig 6). As described above, WPBF and TaSAD recombinant proteins bound to EM1 and DOF2 cis-motifs, whereas no band shift was observed when SPA was incubated with either of them. Interestingly though, the retarded band of DNA-protein complexes was denser when SPA was in the mixture. This is particularly marked with the EM1 motif in combination with SPA and TaSAD recombinant proteins.

## Discussion

There is good evidence that several aspects of the regulatory network governing GSP synthesis is conserved in cereals, including the key involvement of DOF TFs. For example, orthologs of BPBF identified in maize and rice have been extensively studied [46, 47]. SAD, another DOF TF, regulates hordein gene expression in barley [44]. In this work, we identified SAD wheat orthologs called TaSAD. Phylogenetic relationships indicated that SAD TFs from cereals grouped together whereas *Arabidopsis* DOF TFs are in another group. This latter group seems to be orthologous to the group formed with TaSAD and might reflect the separation between dicotyledon and monocotyledon sequences [61]. WPBF is grouped with its orthologs BPBF and ZmPBF, all involved in regulating cereal storage protein gene expression. Fang et al. [62] have detected 108 wheat TaDOF genes, among which *TaSAD-B* (TraesCS6B02G270100) and *WPBF-D* (TraesCS5D02G161000) genes. The tree *TaSAD* homoeologs and the *OsDof08* genes (*OsDof2* in our study) were found in the same group as well as *WPBF* homoeologs and the *OsDof07* gene (*OsDof3* here). Although barley genes were not included in their analysis, each subgroup was separated from the arabidopsis and maize genes as in our phylogenetic tree.

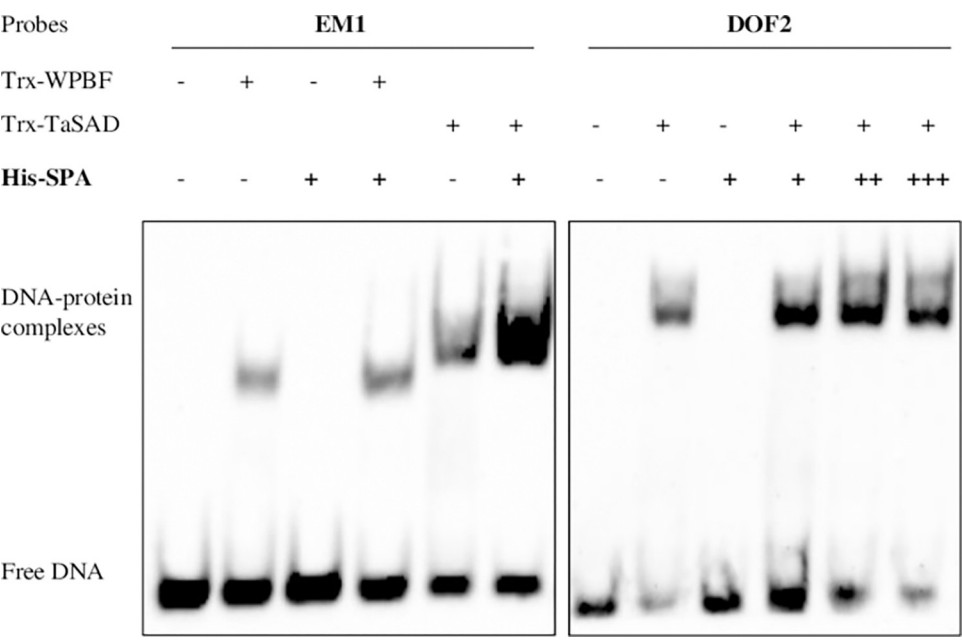

**Fig 6. Electrophoretic mobility shift assays of recombinant TaSAD, WPBF and SPA proteins with oligonucleotide sequences found in the *GluB1-1* (DOF2) and *GluD3* (EM1) promoters.** EMSA of recombinant proteins Trx-TaSAD (370 ng), Trx-WPBF (500 ng) and 625 ng (+), 1250 ng (++), or 2500 ng (+++) of His-SPA with 20 fmol of 20 bp biotin-labeled DNA probes. The probes used were EM1 (-303bp) and DOF2 (-713bp) derived from the *GluD3* and *GluB1-1* gene promoters, respectively.

The involvement of TaSAD and WPBF in regulation of GSP gene transcription was analyzed by *in vitro* and *in vivo* approaches.

### *In vitro* TaSAD and WPBF specifically bound to glutenin promoter DOF cis-motifs

One of the most described cis-motifs in the promoters of genes encoding GSP is the P-box 5′-TGTAAAG-3′, also called EM, which is recognized by DOF TFs. This cis-motif has been identified in promoters of LMW-GS and gliadin genes [18, 20, 63, 64]. In HMW-GS gene promoters, this P-box may be partially conserved [54, 55]. Based on *in vitro* binding assays, TaSAD recombinant protein binds specifically to the DOF and EM cis-motifs in *GluB1-1* and *GluD3* promoters, respectively. It binds more specifically to the core motif AAAG. The ability of WPBF to bind these cis-motifs is consistent with previous works [20, 21]. Shifted bands of DNA-protein complexes were clearly observed for TaSAD or WPBF in the presence of DOF2, DOF3 and EM motifs, but were much fainter in the presence of the DOF1 motif, suggesting different binding affinities for these cis-motifs. The regions flanking the core sequence AAAG of the tested probes were never identical, which may have affected the DNA binding of DOF proteins [65]. TaSAD and WPBF did not seem to differ in their binding affinity. Moreover, TaSAD and WPBF did not bind the P-box motifs described by Norre et al. [55] in the *Glu-D1-2* promoter, or similar ones in the *GluB1-1* promoter. Indeed, the P-box-like cis-motifs in the *GluB1-1* promoter do not contain the AAAG core motif. According to Yanagisawa and Schmidt [65], the AAAG core motif is necessary for the DNA-protein interaction, so this could explain why TaSAD and WPBF did not bind these probes in EMSA.

The DOF domain mediates both DNA-binding and protein-protein interaction. Two shifted bands of DNA-protein complexes were observed with TaSAD and the DOF2 and EM2

motifs suggesting that TaSAD may bind these cis-motifs as dimers. This is surprising as barley SAD was not reported to form homodimers in plant nuclei [45]. Nevertheless, previous works showed that DOF proteins can form homodimers and heterodimers. EMSA results from Yanagisawa [66] showed the formation of homomeric and heteromeric complexes of maize DOF1 and DOF2 proteins. More recently, DNA binding affinities of *Arabidopsis* DOF proteins were assessed [67], who demonstrated that the DOF proteins physically contact each other and bind as a dimer to DNA.

## WPBF and TaSAD regulate glutenin gene expression

*In vitro*, the recognition of DOF and EM motifs of the *GluB1-1* and *GluD3* promoters by TaSAD and WPBF prompted us to investigate whether TaSAD or WPBF function in transient transcriptional regulation of GSP in wheat endosperm. TaSAD and WPBF activated the GUS reporter gene controlled by the *GluB1-1* and *GluD3* promoters. Initial data suggested that WPBF does not independently activate a gene encoding LMW-GS [21], but more recently, WPBF was reported to activate α-gliadin gene expression [23]. Considering promoters differ not only between genes coding for the different GSP classes but also between genes in a given class, the cumulated results suggest that WPBF may be involved in the regulation of all GSP genes. Thus, TaSAD and WPBF are transcriptional activators of GSP gene expression in wheat just as SAD and BPBF are in barley [43, 45]. WPBF activation of *GluB1-1* or *GluD3* expression is higher than with just TaSAD. Moreover, the relative expression of *TaSAD* was low compared to that of *WPBF*. This may indicate that WPBF is the major DOF protein regulator.

When TaSAD was co-bombarded with WPBF, there was no additional transactivation from the *GluB1-1* or *GluD3* promoter above the transactivation by WPBF alone. It is conceivable that this result may be due to all the cis-motifs being occupied by WPBF, which is more abundant at the transcript level than TaSAD. By comparison, Diaz et al. [45] reported that co-bombardment of SAD and PBF had an additive effect on the expression of *Hor2* in transient expression assays in barley, but they detected no interaction between these two DOF proteins.

## Cooperation of SPA with DOF proteins in GSP gene regulation

The presence of the conserved DOF-bZIP cis-regulatory module in glutenin promoters suggests that DOF and bZIP TFs may cooperate. Therefore, we measured the transcriptional activity of *GluB1-1* and *GluD3* in the presence of TaSAD or WPBF with SPA. Transient expression assays provided direct evidence that the activation by either TaSAD or WPBF on glutenin promoters was improved by SPA. Several authors have demonstrated such an additive/synergistic effect between DOF and bZIP TFs. Hwang et al. [68] noticed synergism between maize Opaque2 (O2) and PBF proteins in activating rice globulin and some prolamin (*RP6* and *PG5a*) genes as well as a wheat HMW-GS gene in developing rice endosperm cells. This synergism action was confirmed in maize with two O2 heterodimerizing proteins, OHP1 and OHP2 interacting with PBF to regulate zein synthesis [69]. Transient experiments in rice demonstrated that transactivation effects of RISBZ1 with RPBF were much higher than the additive effect of each one individually, indicating a synergistic interaction between RISBZ1 and RPBF [47]. Experiments of Zhu et al. [22] have confirmed that over-expression of *TaPBF-D* promoted the accumulation of glutenin and an up-regulation of *TaSPA*, suggesting a cooperation of TaSPA with TaPBF to regulate the *Glu-1* genes.

Physical interactions between bZIP and DOF proteins, which recognize the GLM and P-box motifs, respectively, have also been reported. Vicente-Carbajosa et al. [46] showed in a pull-down experiment that maize PBF interacts *in vitro* with O2. More pull-down experiments confirmed that O2 heterodimerizing proteins can interact with PBF [69]. Recently, physical

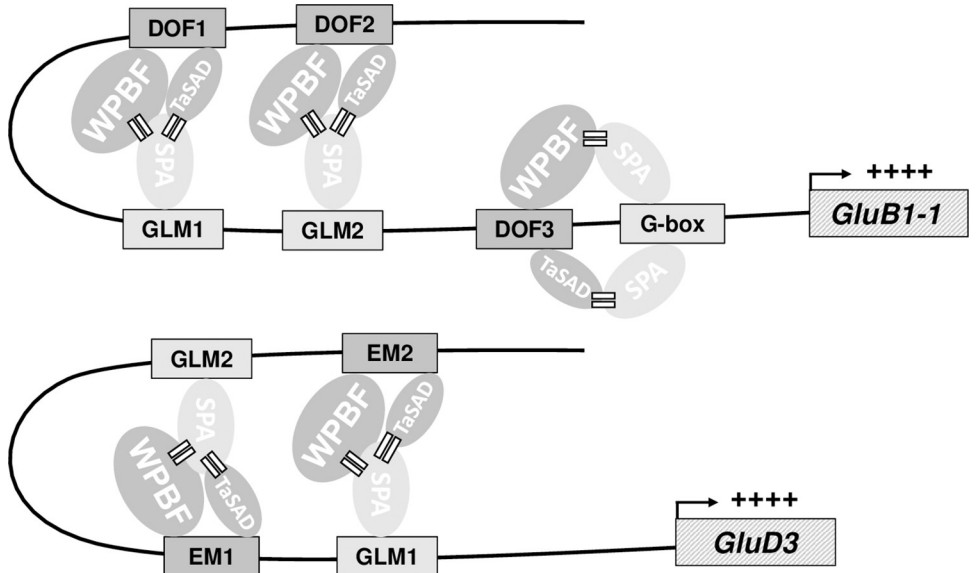

**Fig 7. Proposed model of the transcriptional regulation of the *GluB1-1* and *GluD3* genes in wheat endosperm mediated by SPA interacting with WPBF and TaSAD.** Proteins (ellipses) are shown binding to specific DNA motifs (shaded rectangles) in promoters upstream of glutenin genes (hatched rectangles). The size of the ellipses reflects the expression level of each TF assuming protein abundance correlates with measured relative mRNA abundance. The equal signs suggest possible protein-protein interactions. The plus signs "++++" indicate the activation of the glutenin genes. All name abbreviations are explained in the main text.

interaction between RISBZ1 and RPBF was demonstrated *in vivo* in a bimolecular fluorescence complement assay [70]. Our EMSA experiments suggested that SPA enhanced the binding of TaSAD to EM1 and the binding of WPBF to DOF2 motifs. EMSA has already been used to show the effect of one protein on the DNA binding capacity of another. Indeed, Yanagisawa [66] used EMSA assays to show that binding of maize Dof1 protein to the 35S promoter of the cauliflower mosaic virus was enhanced by the presence of high-mobility-group protein 1. VIVIPAROUS1 (HvVP1) from the barley B3 family decreases the binding affinities of GAMYB and BPBF for their corresponding cis-elements in the promoters of *Hor2* and *Amy6.4* in EMSA assays [71].

Here, we did not directly demonstrate any protein-protein interactions, but transient experiments and EMSA results suggested an active cooperation of SPA with WPBF and TaSAD in glutenin gene expression. Specifically, the maximum expression of *GluB1-1* and *GluD3* genes was obtained when WPBF or TaSAD were co-bombarded with SPA. If mRNA and protein abundance are correlated, we could hypothesize that WPBF, which is more abundant than TaSAD, is the major DOF TF regulator.

The organization of cis-motifs differs between *GluB1-1* and *GluD3* gene promoters, particularly with respect to the EM or DOF cis-motifs, and the GLM and G-box known to bind SPA. For example, EM and GLM form a bipartite box, called the Endosperm Box, which is repeated twice in the *GluD3* promoter [18, 20], but this organization is not found in HMW-GS promoters where DOF1 and DOF2 motifs are located a few nucleotides upstream of GLM and DOF3 near the G-Box [54]. SPA did not bind to EM or DOF cis-motifs but its presence on the GLM and G-box motifs may promote protein-protein interactions between DOF proteins and SPA. Here we propose a model to illustrate the transcriptional regulation of glutenin genes with these three TFs and their corresponding cis-motifs on the promoters (Fig 7). In this model, WPBF and TaSAD regulate the *Glu-B1-1* and *GluD3* promoters in cooperation with SPA. The

DOF TFs bind to the corresponding conserved cis-motifs, while SPA interacts with each type of DOF TF. In this way, the transcriptional activation by the DOF TFs is stabilized. Juhász et al. [64] reported differences between promoters of LMW-GS genes in the number and combination of cis-motifs able to bind DOF TFs or SPA, while Ravel et al. [54] reported such differences for HMW-GS promoters. These differences may explain the differences in LMW-GS or HMW-GS gene expression.

In conclusion, we have identified and characterized TaSAD, the wheat ortholog of barley SAD. Experimental assays show that TaSAD is an activator of *GluB1-1* and *GluD3*, like WPBF. No additive effect between TaSAD and WPBF was observed. The results suggest that GSP activation by TaSAD and WPBF might be enhanced through their cooperation with SPA. More work is needed to fully describe the functionality of these DOF proteins and in particular whether they interact with other TFs, such as Gamyb, which interacts with BPBF and SAD in barley [45, 49]. Non-transcriptional protein regulators may be involved too, such as TaQM, which has an additive effect in combination with WPBF on the expression of gliadin genes [23]. In addition, post-translational modifications of TFs, in particular phosphorylation/ dephosphorylation, may be involved in this complex regulatory network by activating or inactivating DNA-binding activities. For instance, it has been reported in maize that only the hypo-phosphorylated form of O2 and the phosphorylated ZmPBF have DNA-binding activity [72, 73]. Finding TaSAD provides a more complete understanding of the regulatory network of GSP synthesis in wheat, which could indicate ways to modify GSP composition to obtain the rheological quality suited for a given process.

## Supporting information

**S1 Fig.** Expression and purification of recombinant proteins Trx-TaSAD (a) and Trx-WPBF (b). Crude extracts from uninduced or induced bacteria harboring the pET32Trx-WPBF and pET32Trx-TaSAD expression vectors and the proteins not retained (FlowThrough) or eluted during recombinant protein purification resolved by electrophoresis through a 10% SDS-poly-acrylamide gel. MW: molecular weight markers, sizes are in kilodaltons (kDa).
(TIF)

**S2 Fig.** *In silico* **annotation of the** *GluB1-1* **promoter and electrophoretic mobility shift assays of recombinant Trx-WPBF and Trx-TaSAD proteins with oligonucleotides derived from the** *GluB1-1* **and** *GluD3* **gene promoter.** (a) Schematic representation of *GluB1-1* promoter. The TATA box and nucleotide positions relative to the start codon are indicated. Putative cis-motifs, DOF1 (-742bp), DOF2 (-713bp) and DOF3 (-345bp), Pb3-like (-420bp), Pb2 (-314bp) and Pb1-like (-244bp), G-box (−280bp), GATA box (−661bp, −640bp, −636bp) and AACA motif (−580bp), GLM1 and GLM2 (−652bp and -631bp) are shown. (b) EMSA of recombinant Trx-WPBF (500 ng) and Trx-TaSAD (370 ng) proteins with the 22 bp biotin-labeled Pb3-like (-420bp) and EM2 (-337bp) probes derived from the *GluB1-1* and *GluD3* promoters respectively.
(TIF)

**S1 Table. List of DOF proteins used in phylogenetic tree analysis.**
(DOCX)

**S2 Table. Sequences of the primers used for qRT-PCR in cv. NB1.** (a) Housekeeping gene primers used for normalization of relative gene expression.
(DOCX)

**S3 Table. Sequences of Prolamin-box (Pb), DOF, EM and mutated versions (*dof* and *em*) probes used in EMSA.** Probe sequences are shown with the cis-motifs in bold and mutated nucleotides in lowercase.
(DOCX)

## Acknowledgments

We thank the experimental infrastructure VégéPôle from the UMR GDEC and more particularly Richard Blanc and Michael Denefle for their help in plant cultivation. We also thank Rachel Carol from Bioscience Editing for English improvement of the manuscript.

## Author Contributions

**Conceptualization:** Marielle Merlino, Pierre Martre, Catherine Ravel, Julie Boudet.

**Data curation:** Marielle Merlino.

**Formal analysis:** Marielle Merlino.

**Funding acquisition:** Pierre Martre, Catherine Ravel, Julie Boudet.

**Investigation:** Marielle Merlino, Jean-Charles Gaudin, Mireille Dardevet.

**Supervision:** Pierre Martre, Catherine Ravel, Julie Boudet.

**Writing – original draft:** Marielle Merlino.

**Writing – review & editing:** Marielle Merlino, Jean-Charles Gaudin, Mireille Dardevet, Pierre Martre, Catherine Ravel, Julie Boudet.

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
