## [Decision Letter · Decision Letter 0]

27 Mar 2023

PONE-D-23-01362Wheat DOF transcription factors TaSAD and WPBF regulate glutenin gene expression in cooperation with SPAPLOS ONE

Dear Dr. Boudet,

Thank you for submitting your manuscript to PLOS ONE. After careful consideration, we feel that it has merit but does not fully meet PLOS ONE’s publication criteria as it currently stands. Therefore, we invite you to submit a revised version of the manuscript that addresses the points raised during the review process.

We look forward to receiving your revised manuscript.

Kind regards,

Shailender Kumar Verma, Ph.D.

Academic Editor

PLOS ONE

“We thank the experimental infrastructure VégéPôle from the UMR GDEC and more particularly Richard Blanc and Michael Denefle for their help in plant cultivation. We also thank Rachel Carol from Bioscience Editing for English improvement of the manuscript. This work was supported by the French National Research Agency (ANR) and France AgriMer in the framework of the Investments for the Future BreedWheat project (ANR-10-BTBR-03).”

‘This work was supported by the French National Research Agency (ANR) and France AgriMer in the framework of the Investments for the Future BreedWheat project (ANR-10-BTBR-03).

No, the funders had no role in study design, data collection and analysis, decision to publish, or preparation of the manuscript.”

Reviewers' comments:

Reviewer's Responses to Questions

**Comments to the Author**

1. Is the manuscript technically sound, and do the data support the conclusions?

Reviewer #1: Yes

Reviewer #2: Yes

2. Has the statistical analysis been performed appropriately and rigorously? 

Reviewer #1: Yes

Reviewer #2: Yes

3. Have the authors made all data underlying the findings in their manuscript fully available?

Reviewer #1: Yes

Reviewer #2: Yes

4. Is the manuscript presented in an intelligible fashion and written in standard English?

Reviewer #1: Yes

Reviewer #2: Yes

5. Review Comments to the Author

Reviewer #1: 1. The manuscript work is novel with regard to the study of TaSAD gene and the cumulative effect along with WPBF in regulation of glutein gene expression in cooperation with SPA in wheat.

2. However, in the introduction references of the Dof regulating PBF, SAD and carbon & nitrogen metabolism reported in finger millet or other cereal crops should be cited.  

3. References are not arranged alphabetically, and the alignment is not justified.

4. The conclusions are well explained. The author has efficiently provided the gene construct model and data has been statistically analyzed.

5. The manuscript adheres to the guidelines and may be considered for publication if citation in the introduction and proper references and in appropriate style as per instruction to authors is added.

Reviewer #2: Manuscript ID# PONE-D-23-01362 entitled “Wheat DOF transcription factors TaSAD and WPBF regulate glutenin gene expression in cooperation with SPA” submitted to PLOS ONE. In this article, the authors identified SAD wheat orthologs called TaSAD. EMSA and transient expression assays on immature wheat endosperms were performed to determine its DNA binding and regulatory activities on glutenin gene expression. TaSAD function was also characterized in combination with WPBF and SPA proteins.

Major revisions:

(1) Please check that the significance of differences in Figure 5 is correctly labelled. In addition, the whole figure notes are too cumbersome and not indicate whether duplication is done.

(2) The discussion section does not go far enough, please add some depth and focus on the latest research and provide more novel and unique viewpoints about the research in the field.

Minor revisions:

(1) Please check that the format of the references is correct and consistent. For instance, the 45th, “Oryza sativa L.” should be “Oryza sativa L.”.

(2) Please check the correct writing in the article. In the article, “WPBF” is not italic.

(3) Please provide clearer images in the article.

6. PLOS authors have the option to publish the peer review history of their article (what does this mean?). If published, this will include your full peer review and any attached files.

Reviewer #1: **Yes: **Prof. Anil Kumar Gupta

Reviewer #2: **Yes: **Dongfang Ma

---

## [Author Response · Author response to Decision Letter 0]

11 May 2023

Reviewer #1: 

1. The manuscript work is novel with regard to the study of TaSAD gene and the cumulative effect along with WPBF in regulation of glutein gene expression in cooperation with SPA in wheat.

2. However, in the introduction references of the Dof regulating PBF, SAD and carbon & nitrogen metabolism reported in finger millet or other cereal crops should be cited.

Response:

In the introduction section, recent references about grain storage protein synthesis regulation in finger millet [12], in maize [13], in rice [14] and comparison between monocots and dicots [15] have been added (lines 61 to 62). These illustrate a conserved regulatory network in cereals and dicots.

The paragraph on the biological functions of DOFs (lines 85 to 89) has been expanded and several references have been added [31-33; 34-37; 38-41], as an example, the recent review about DOF proteins by Zou et al., 2023 (line 86). Moreover, references about finger millet DOF TFs involved in carbon and nitrogen assimilation [35] and in carbohydrate metabolism [39] (lines.88 to 89) have been added.

To illustrate DOF and bZIP protein interactions, a reference about finger millet DOF and O2 proteins has been added [48] (lines 112 to 114). This work suggested a possible interaction between EcO2 and EcDOF proteins and this heterodimer could have a high binding activity on GSP gene promoters.

3. References are not arranged alphabetically, and the alignment is not justified.

Response:

According to PLOS ONE’s instructions, we have arranged the references in order of appearance in the text. In the revised version, the alignment of references is justified and the style preference is “PLOSONE” from the Zotero software. 

4. The conclusions are well explained. The author has efficiently provided the gene construct model and data has been statistically analyzed.

Response:

Thank you for your positive comments on our manuscript.

5. The manuscript adheres to the guidelines and may be considered for publication if citation in the introduction and proper references and in appropriate style as per instruction to authors is added.

Reviewer #2: 

Manuscript ID# PONE-D-23-01362 entitled “Wheat DOF transcription factors TaSAD and WPBF regulate glutenin gene expression in cooperation with SPA” submitted to PLOS ONE. In this article, the authors identified SAD wheat orthologs called TaSAD. EMSA and transient expression assays on immature wheat endosperms were performed to determine its DNA binding and regulatory activities on glutenin gene expression. TaSAD function was also characterized in combination with WPBF and SPA proteins.

Major revisions:

(1) Please check that the significance of differences in Figure 5 is correctly labelled. In addition, the whole figure notes are too cumbersome and not indicate whether duplication is done.

Response:

We checked the significance of differences in Figure and it is correctly labelled.

The caption of Fig 5 has been simplified. The constructs used are the same as in Fig 4 (except for the p-UbiSPA). The values of the normalized GUS expression from the reporters without effector are identical to those in Fig 4. So, this is specified in the new version of the caption. 

(2) The discussion section does not go far enough, please add some depth and focus on the latest research and provide more novel and unique viewpoints about the research in the field.

Response:

We added more recent works to further support our results. In particular, to support our phylogenetic analysis, the work of Fang et al., 2020 [63] has been added (lines 454 to 460). They detected 108 wheat TaDOF genes among which TaSAD and WPBF. As in our analysis, these were separated from Arabidopsis and maize DOF.

More recent works have been added to support the fact that SPA cooperated with DOF proteins in GSP gene regulation [70] (lines 531 to 533) and [71] (lines 536 to 539) 

Minor revisions:

(1) Please check that the format of the references is correct and consistent. For instance, the 45th, “Oryza sativa L.” should be “Oryza sativa L.”.

Response:

We have checked the format of all references in the revised version. 

(2) Please check the correct writing in the article. In the article, “WPBF” is not italic.

Response:

We have checked the correct writing of the names of gene (italic) and those of protein (no italic). This is the list of modified writing format: lines 162 and 163 WPBF and TaSAD transformed in italic

(3) Please provide clearer images in the article.

Response:

All the figures and supplemental figures were uploaded in PACE which converts figures to TIF format, resizes, and renames them according to the file naming conventions. We provide these new figures processed with PACE.

---

## [Decision Letter · Decision Letter 1]

12 Jun 2023

Wheat DOF transcription factors TaSAD and WPBF regulate glutenin gene expression in cooperation with SPA

PONE-D-23-01362R1

Dear Dr. Boudet,

We’re pleased to inform you that your manuscript has been judged scientifically suitable for publication and will be formally accepted for publication once it meets all outstanding technical requirements.

Kind regards,

Shailender Kumar Verma, Ph.D.

Academic Editor

PLOS ONE

Additional Editor Comments (optional):

Reviewers' comments:

Reviewer's Responses to Questions

**Comments to the Author**

1. If the authors have adequately addressed your comments raised in a previous round of review and you feel that this manuscript is now acceptable for publication, you may indicate that here to bypass the “Comments to the Author” section, enter your conflict of interest statement in the “Confidential to Editor” section, and submit your "Accept" recommendation.

Reviewer #2: All comments have been addressed

2. Is the manuscript technically sound, and do the data support the conclusions?

Reviewer #2: Yes

3. Has the statistical analysis been performed appropriately and rigorously? 

Reviewer #2: Yes

4. Have the authors made all data underlying the findings in their manuscript fully available?

Reviewer #2: Yes

5. Is the manuscript presented in an intelligible fashion and written in standard English?

Reviewer #2: Yes

6. Review Comments to the Author

Reviewer #2: Grain storage proteins (GSPs) quantity and composition determine the end-use value of wheat flour. The researcher TaSAD and WPBF activate GSP gene expression.Moreover, co-bombardment of Storage Protein Activator (SPA) with WPBF or TaSAD had an additive effect on the expression of GSP genes, possibly through conserved cooperative protein-protein interactions.

The manuscript can be accept for publication.

7. PLOS authors have the option to publish the peer review history of their article (what does this mean?). If published, this will include your full peer review and any attached files.

Reviewer #2: No
